# Geochemical Constraints on Mantle Melting and Magma Genesis at Pohnpei Island, Micronesia

**Tong Zong** [1] , **Zheng-Gang Li** [2,*] , **Yan-Hui Dong** [2], **Xu-Ping Li** [1] , **Ji-Hao Zhu** [2], **Ling Chen** [2] **and Ji-Qiang Liu** [2]

1   Shandong Provincial Key Laboratory of Depositional Mineralization & Sedimentary Minerals, Shandong University of Science and Technology, Qingdao 266590, China; zongtong2019@sdust.edu.cn (T.Z.); lixuping@sdust.edu.cn (X.-P.L.)
2   Key Laboratory of Submarine Geosciences, Second Institute of Oceanography, Ministry of Natural Resources, Hangzhou 310012, China; dongyh@sio.org.cn (Y.-H.D.); zhujh@sio.org.cn (J.-H.Z.); chenling@sio.org.cn (L.C.); liujq@sio.org.cn (J.-Q.L.)
*   Correspondence: lizg@sio.org.cn

**Abstract:** The lithospheric mantle is of paramount importance in controlling the chemical composition of ocean island basalts (OIBs), influencing partial melting and magma evolution processes. To improve the understanding of these processes, the pressure–temperature conditions of mantle melting were investigated, and liquid lines of descent were modelled for OIBs on Pohnpei Island. The studied basaltic samples are alkalic, and can be classified as $SiO_2$-undersaturated or $SiO_2$-saturated series rocks, with the former having higher $TiO_2$ and $FeO^T$ contents but with no distinct trace-element composition, suggesting melting of a compositionally homogenous mantle source at varying depths. Both series underwent sequential crystallization of olivine, clinopyroxene, Fe–Ti oxides, and minor plagioclase and alkali feldspar. Early magnetite crystallization resulted from initially high $FeO^T$ contents and oxygen fugacity, and late feldspar crystallization was due to initially low $Al_2O_3$ contents and alkali enrichment of the evolved magma. The Pohnpei lavas formed at estimated mantle-melting temperatures of 1486–1626 °C (average 1557 ± 43 °C, 1σ), and pressures of 2.9–5.1 GPa (average 3.8 ± 0.7 GPa), with the $SiO_2$-undersaturated series forming at higher melting temperatures and pressures. Trace-element compositions further suggest that garnet rather than spinel was a residual phase in the mantle source during the melting process. Compared with the Hawaiian and Louisville seamount chains, Pohnpei Island underwent much lower degrees of mantle melting at greater depth, possibly due to a thicker lithosphere.

**Keywords:** lithospheric mantle; mantle melting; fractional crystallization; OIB-type lavas; Pohnpei Island

## 1. Introduction

Ocean island basalts (OIBs) are widely considered to be produced by deep-seated thermal mantle plumes from the core–mantle boundary [1] or a shallower mantle transition zone [2], and their chemical compositions have commonly been used to constrain the chemistry of mantle sources. However, determination of the chemical composition of the mantle source is not straightforward, due to overprinting by mantle melting [3], melt–lithosphere interaction [4–6], and fractional crystallization [7]. Variations in lithospheric thickness are thus of paramount importance in controlling the chemical composition of OIBs; although, this is not well understood.

The Caroline Islands in Micronesia were formed on a thick, ancient Pacific lithosphere at 162–153 Ma [8], and are associated with the Caroline hotspot [9–13]. Three of these volcanic

islands—Chuuk, Pohnpei, and Kosrae—display progressive age variations, and are spatiotemporally related to the Caroline Plateau [13]. Jackson et al. [10] suggested that the Caroline Islands were formed by partial melting of a compositionally homogeneous mantle source, with a small contribution of recycled oceanic crust. However, the effect of thick lithospheric mantle on the chemical composition of Caroline Island lavas has not yet been explored. In this study, we undertook whole-rock major- and trace-element, and in situ chemical analyses of volcanic lavas from Pohnpei Island, with the aim of exploring the effects of fractional crystallization and mantle melting on their chemical compositions.

## 2. Geological Background

The three volcanic islands, Pohnpei, Chuuk, and Kosrae, are associated with the Caroline Seamount Chain connected to the western Caroline Plateau (Figure 1), which developed at 23.9–8.13 Ma [13–15] after formation of the Caroline Plateau, and their ages decrease from west to east (since the middle Cenozoic): Chuuk, 14.8–4.3 Ma; Pohnpei, 8.7–0.92 Ma; and Kosrea, 2–1 Ma [16]. They are considered to be derived from a common mantle source related to the deep-rooted Caroline hotspot, which erupted through an old (162–153 Ma) portion of the Pacific Ocean crust with a relatively thick lithosphere [8–11]. The estimated lithosphere–asthenosphere boundary depth for these islands is ~96 km [17], thicker than those of the Caroline Plateau [13] and Hawaiian Islands (88–90 km) [17]. The $^3$He/$^4$He ratios (7.6–12.8 Ra) of basalts from these islands indicate a deep mantle source [10,13,16], possibly related to recycled oceanic components [10]. However, Caroline lavas do not have the high $^3$He/$^4$He compositions that are characteristic of Hawaii, Iceland, and Samoa lavas among others, possibly because they are more degassed, or higher U–Th concentrations were maintained in their mantle source over geologic time [10]. The absence of newer volcanism near Kosrae, the youngest island, suggests the activity of the Caroline hotspot has decreased or, more likely, ceased [13,16].

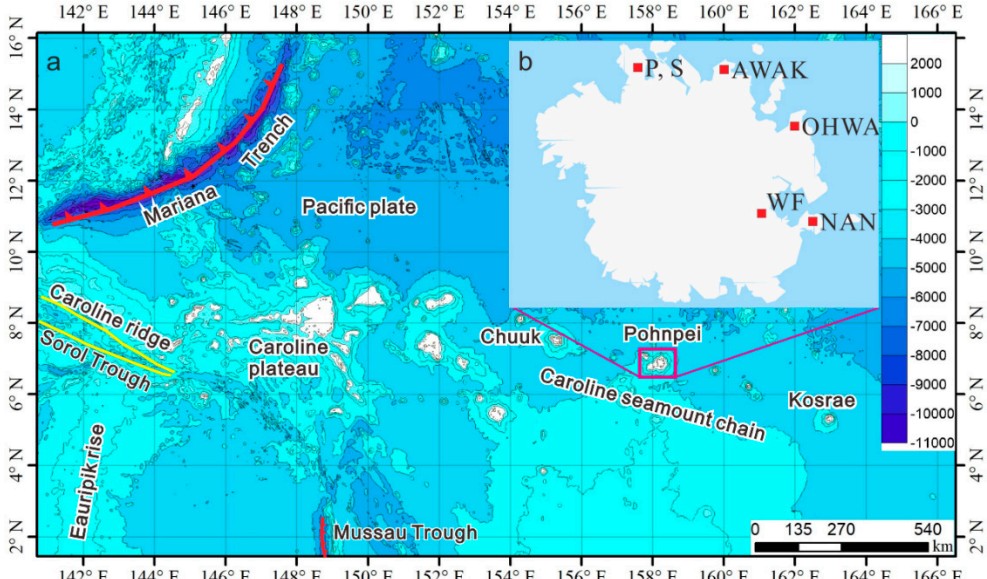

**Figure 1.** (**a**) Bathymetric map of the Caroline Seamount Chain showing the major tectonic features of the seamount chain, Caroline Plateau, and adjacent areas. (**b**) The sampling locations on the Pohnpei Island. The abbreviations P, S, AWAK, OHWA, WF, and NAN represent sampling stations, as described in Table 1.

Pohnpei Island (6°54′N lat, 158°14′E long) is roughly circular in shape, almost completely surrounded by a barrier reef, and is characterized by extensive dike systems and columnar jointing [14–16]. Lavas exposed on Pohnpei comprise alkali basalts, trachytes, hawaiites, ankramites, and nephlinites/basanites [16]. They can be subdivided into three stages based on chemical composition: the main lava, transitional lava, and nephelinite/basanite series [14,18]. The former two

represent shield-building stages of volcanism, and the nephelinite/basanite series represents overlying post-shield deposits [14]. The main lava series comprises basalt, hawaiite, mugearite, and trachyte, with hawaiite being the most abundant and containing phenocrysts of olivine, clinopyroxene, and rare magnetite [16]. The transitional lava series contains olivine, clinopyroxene, and plagioclase phenocrysts. The nephelinite/basanite series is $SiO_2$-undersaturated and distinguished from the main lava series by a narrow range of higher more- to less-incompatible element ratios [16].

**Table 1.** Major- and trace-element (ppm) contents of lavas from Pohnpei Island. (LOI—Loss on ignition).

| Sample No. | P-1 | P-2 | WF-1 | WF-2 | WF-3 | NAN-1 | NAN-2 | NAN-3 | S1 | S3 |
|---|---|---|---|---|---|---|---|---|---|---|
| $SiO_2$ (wt%) | 45.64 | 44.85 | 44.34 | 45.53 | 46.42 | 43.59 | 42.78 | 43.99 | 44.55 | 48.24 |
| $TiO_2$ | 3.10 | 3.02 | 3.50 | 2.33 | 3.19 | 3.50 | 3.68 | 3.66 | 3.43 | 2.56 |
| $Al_2O_3$ | 13.21 | 12.92 | 11.90 | 12.31 | 15.93 | 11.36 | 12.62 | 13.17 | 14.84 | 15.77 |
| $Fe_2O_3$ | 13.46 | 13.30 | 14.41 | 13.25 | 11.15 | 14.60 | 14.79 | 14.03 | 13.30 | 11.66 |
| MnO | 0.17 | 0.17 | 0.18 | 0.17 | 0.18 | 0.17 | 0.17 | 0.17 | 0.19 | 0.18 |
| MgO | 8.55 | 8.77 | 10.84 | 11.95 | 4.67 | 11.09 | 9.55 | 8.19 | 6.16 | 5.46 |
| CaO | 9.88 | 9.84 | 11.20 | 10.91 | 8.86 | 11.31 | 10.55 | 10.71 | 9.93 | 8.57 |
| $Na_2O$ | 2.95 | 2.79 | 2.26 | 2.95 | 4.30 | 2.53 | 2.94 | 2.92 | 4.43 | 3.50 |
| $K_2O$ | 1.12 | 1.09 | 1.16 | 0.56 | 1.86 | 0.91 | 1.07 | 1.23 | 1.66 | 1.47 |
| $P_2O_5$ | 0.57 | 0.56 | 0.53 | 0.57 | 0.80 | 0.49 | 0.57 | 0.63 | 0.99 | 0.77 |
| LOI | 0.98 | 1.28 | 0.64 | 0.05 | 0.26 | 0.05 | 0.00 | 0.43 | 0.22 | 1.52 |
| Sum | 99.64 | 98.59 | 100.95 | 100.59 | 97.61 | 99.60 | 98.71 | 99.13 | 99.69 | 99.70 |
| Li (ppm) | 5.46 | 5.55 | 5.67 | 5.42 | 7.14 | 5.07 | 5.54 | 6.09 | 8.09 | 6.64 |
| Sc | 24.87 | 25.26 | 29.80 | 27.10 | 12.33 | 31.92 | 25.27 | 27.52 | 17.80 | 16.20 |
| V | 250.7 | 254.5 | 730.0 | 665.3 | 395.5 | 847.9 | 329.3 | 378.1 | 252.0 | 191.0 |
| Cr | 313.3 | 319.9 | 401.8 | 435.2 | 28.6 | 456.5 | 342.0 | 240.2 | 110.0 | 124.0 |
| Co | 65.57 | 53.31 | 66.98 | 64.28 | 71.86 | 65.16 | 63.33 | 56.14 | 70.20 | 57.30 |
| Ni | 162.0 | 166.6 | 128.6 | 131.5 | 25.4 | 108.0 | 171.0 | 139.4 | 74.2 | 74.0 |
| Cu | 62.68 | 58.02 | 53.10 | 69.19 | 39.51 | 90.68 | 56.46 | 76.79 | 44.60 | 34.20 |
| Zn | 137.0 | 133.1 | 136.0 | 121.2 | 126.1 | 130.3 | 142.3 | 143.9 | 156.0 | 137.0 |
| Rb | 49.81 | 49.37 | 41.65 | 24.98 | 40.28 | 42.39 | 52.87 | 38.07 | 46.70 | 31.50 |
| Sr | 674.2 | 683.2 | 847.2 | 612.9 | 913.2 | 602.9 | 729.2 | 796.2 | 1090 | 775.0 |
| Y | 27.42 | 27.00 | 26.21 | 25.07 | 31.51 | 24.49 | 27.21 | 31.04 | 36.20 | 33.30 |
| Zr | 257.3 | 257.3 | 275.0 | 178.6 | 361.6 | 217.4 | 240.5 | 292.0 | 426.0 | 409.0 |
| Nb | 38.37 | 38.90 | 48.89 | 43.38 | 62.22 | 35.42 | 43.78 | 46.49 | 81.80 | 60.40 |
| Cs | 0.36 | 0.43 | 0.48 | 0.27 | 0.70 | 0.36 | 0.40 | 0.42 | 0.45 | 0.20 |
| Ba | 334.7 | 346.0 | 432.9 | 392.6 | 575.2 | 328.0 | 411.5 | 410.9 | 694.0 | 476.0 |
| La | 33.52 | 32.83 | 40.01 | 43.32 | 53.20 | 30.05 | 34.47 | 40.47 | 60.70 | 48.70 |
| Ce | 70.85 | 70.13 | 80.18 | 80.58 | 106.19 | 63.76 | 71.93 | 84.89 | 126.0 | 104.0 |
| Pr | 9.57 | 9.48 | 10.43 | 10.03 | 13.27 | 8.67 | 9.77 | 11.50 | 15.20 | 12.60 |
| Nd | 40.73 | 40.14 | 43.12 | 40.18 | 53.96 | 37.32 | 41.48 | 48.18 | 60.90 | 50.90 |
| Sm | 8.36 | 8.22 | 8.59 | 7.67 | 10.32 | 7.76 | 8.66 | 9.96 | 12.30 | 10.30 |
| Eu | 2.63 | 2.61 | 2.70 | 2.39 | 3.23 | 2.49 | 2.78 | 3.11 | 3.83 | 3.19 |
| Gd | 7.28 | 7.22 | 7.48 | 6.82 | 8.98 | 6.86 | 7.52 | 8.58 | 11.50 | 9.58 |
| Tb | 1.12 | 1.12 | 1.13 | 1.03 | 1.35 | 1.06 | 1.15 | 1.31 | 1.54 | 1.32 |
| Dy | 5.91 | 5.86 | 5.77 | 5.34 | 6.88 | 5.48 | 5.91 | 6.76 | 7.55 | 6.73 |
| Ho | 1.02 | 1.01 | 0.99 | 0.94 | 1.19 | 0.92 | 1.01 | 1.16 | 1.32 | 1.22 |
| Er | 2.44 | 2.43 | 2.36 | 2.25 | 2.85 | 2.18 | 2.36 | 2.74 | 3.16 | 3.04 |
| Tm | 0.39 | 0.38 | 0.37 | 0.36 | 0.46 | 0.34 | 0.37 | 0.43 | 0.41 | 0.42 |
| Yb | 2.21 | 2.18 | 2.12 | 2.10 | 2.58 | 1.88 | 2.07 | 2.43 | 2.29 | 2.39 |
| Lu | 0.29 | 0.29 | 0.27 | 0.28 | 0.34 | 0.25 | 0.27 | 0.32 | 0.33 | 0.35 |
| Hf | 6.15 | 6.24 | 6.42 | 4.35 | 8.03 | 5.43 | 5.86 | 6.94 | 8.71 | 8.65 |
| Ta | 2.42 | 2.46 | 3.03 | 2.42 | 3.93 | 2.19 | 2.70 | 3.01 | 5.32 | 4.04 |
| Pb | 3.53 | 3.43 | 4.09 | 3.22 | 6.06 | 3.08 | 3.38 | 3.94 | 4.34 | 3.12 |
| Th | 2.30 | 2.35 | 3.62 | 4.46 | 5.51 | 2.06 | 2.85 | 3.37 | 6.97 | 4.99 |
| U | 0.90 | 0.89 | 1.09 | 1.12 | 1.50 | 0.85 | 0.94 | 1.09 | 1.86 | 1.40 |

Table 1. *Cont.*

| Sample No. | S4-2 | S5 | S6 | S7 | S8 | AWAK-1 | AWAK-2-1 | AWAK-2-2 | OHWA-1-1 | OHWA-1-2 |
|---|---|---|---|---|---|---|---|---|---|---|
| $SiO_2$ (wt%) | 46.12 | 46.34 | 46.19 | 45.28 | 45.19 | 45.86 | 45.76 | 45.95 | 47.94 | 48.07 |
| $TiO_2$ | 3.09 | 3.17 | 3.18 | 3.03 | 3.01 | 3.32 | 3.46 | 3.54 | 2.94 | 3.01 |
| $Al_2O_3$ | 14.20 | 14.42 | 13.91 | 13.48 | 13.52 | 15.01 | 12.91 | 13.02 | 16.28 | 16.58 |
| $Fe_2O_3$ | 12.97 | 13.20 | 13.15 | 13.31 | 13.24 | 12.86 | 13.35 | 13.55 | 12.14 | 12.30 |
| MnO | 0.17 | 0.17 | 0.17 | 0.17 | 0.17 | 0.19 | 0.18 | 0.18 | 0.17 | 0.18 |
| MgO | 6.99 | 6.87 | 7.35 | 8.98 | 8.86 | 6.19 | 6.50 | 6.55 | 4.46 | 4.70 |
| CaO | 9.82 | 9.56 | 9.84 | 10.31 | 9.93 | 9.86 | 10.83 | 10.50 | 8.22 | 8.26 |
| $Na_2O$ | 2.91 | 3.34 | 3.10 | 2.67 | 2.83 | 3.21 | 2.34 | 2.39 | 3.78 | 3.79 |
| $K_2O$ | 1.21 | 1.22 | 1.22 | 1.07 | 1.19 | 1.22 | 1.25 | 1.26 | 1.46 | 1.46 |
| $P_2O_5$ | 0.64 | 0.64 | 0.64 | 0.54 | 0.55 | 0.79 | 0.44 | 0.43 | 0.93 | 0.94 |
| LOI | 1.58 | 0.97 | 1.46 | 0.76 | 1.01 | 1.53 | 2.43 | 2.29 | 0.87 | 0.75 |
| Sum | 99.69 | 99.92 | 100.20 | 99.59 | 99.49 | 100.04 | 99.45 | 99.65 | 99.19 | 100.04 |
| Li (ppm) | 6.08 | 6.12 | 6.25 | 4.99 | 5.34 | 6.32 | 2.70 | 2.74 | 5.22 | 5.91 |
| Sc | 20.90 | 21.10 | 22.00 | 23.40 | 22.10 | 17.50 | 29.70 | 29.60 | 13.60 | 13.70 |
| V | 257.0 | 256.0 | 265.0 | 271.0 | 255.0 | 227.0 | 311.0 | 312.0 | 173.0 | 173.0 |
| Cr | 197.0 | 179.0 | 222.0 | 336.0 | 316.0 | 97.8 | 129.0 | 126.0 | 40.8 | 40.7 |
| Co | 62.70 | 63.70 | 72.50 | 76.20 | 64.50 | 55.20 | 49.60 | 54.20 | 38.90 | 37.50 |
| Ni | 110.0 | 106.0 | 124.0 | 172.0 | 160.0 | 69.6 | 78.7 | 77.6 | 22.8 | 23.3 |
| Cu | 49.30 | 50.70 | 49.90 | 51.80 | 49.80 | 44.30 | 88.60 | 90.40 | 21.90 | 24.50 |
| Zn | 144.0 | 133.0 | 131.0 | 123.0 | 119.0 | 139.0 | 117.0 | 115.0 | 133.0 | 133.0 |
| Rb | 24.20 | 26.00 | 23.00 | 24.10 | 26.20 | 35.10 | 23.70 | 23.50 | 23.60 | 24.20 |
| Sr | 866.0 | 634.0 | 725.0 | 707.0 | 665.0 | 942.0 | 438.0 | 435.0 | 819.0 | 793.0 |
| Y | 31.20 | 31.70 | 30.90 | 26.80 | 26.70 | 34.00 | 33.60 | 32.70 | 32.60 | 33.20 |
| Zr | 317.0 | 326.0 | 306.0 | 263.0 | 260.0 | 368.0 | 281.0 | 276.0 | 335.0 | 336.0 |
| Nb | 49.00 | 50.40 | 48.60 | 41.40 | 39.70 | 66.40 | 31.20 | 30.60 | 51.90 | 52.10 |
| Cs | 0.31 | 0.35 | 0.33 | 0.28 | 0.30 | 0.64 | 0.06 | 0.06 | 0.13 | 0.15 |
| Ba | 401.0 | 386.0 | 383.0 | 346.0 | 355.0 | 587.0 | 211.0 | 210.0 | 442.0 | 435.0 |
| La | 38.10 | 38.40 | 37.40 | 31.60 | 32.20 | 52.30 | 25.60 | 25.10 | 42.00 | 42.20 |
| Ce | 83.10 | 83.00 | 81.30 | 68.40 | 69.40 | 109.0 | 61.10 | 59.90 | 92.90 | 93.90 |
| Pr | 10.30 | 10.40 | 10.10 | 8.70 | 8.94 | 13.30 | 8.44 | 8.34 | 12.20 | 12.30 |
| Nd | 42.80 | 43.20 | 42.20 | 36.60 | 37.90 | 54.40 | 37.20 | 36.70 | 51.80 | 51.90 |
| Sm | 9.12 | 9.21 | 9.03 | 7.97 | 8.15 | 11.00 | 8.74 | 8.62 | 10.70 | 10.80 |
| Eu | 2.87 | 2.91 | 2.88 | 2.57 | 2.63 | 3.51 | 2.75 | 2.71 | 3.32 | 3.35 |
| Gd | 8.78 | 8.88 | 8.77 | 7.67 | 7.51 | 9.97 | 8.28 | 8.17 | 9.41 | 9.47 |
| Tb | 1.24 | 1.26 | 1.23 | 1.08 | 1.09 | 1.42 | 1.29 | 1.26 | 1.35 | 1.36 |
| Dy | 6.36 | 6.49 | 6.34 | 5.52 | 5.69 | 7.30 | 6.91 | 6.90 | 7.00 | 7.01 |
| Ho | 1.15 | 1.17 | 1.14 | 0.99 | 1.02 | 1.29 | 1.29 | 1.28 | 1.25 | 1.27 |
| Er | 2.82 | 2.90 | 2.81 | 2.42 | 2.50 | 3.15 | 3.29 | 3.26 | 3.12 | 3.15 |
| Tm | 0.38 | 0.39 | 0.38 | 0.32 | 0.33 | 0.42 | 0.45 | 0.44 | 0.42 | 0.42 |
| Yb | 2.16 | 2.22 | 2.12 | 1.82 | 1.88 | 2.32 | 2.59 | 2.56 | 2.40 | 2.42 |
| Lu | 0.31 | 0.32 | 0.30 | 0.26 | 0.28 | 0.34 | 0.38 | 0.37 | 0.35 | 0.35 |
| Hf | 7.06 | 7.22 | 6.84 | 5.95 | 6.42 | 8.66 | 7.16 | 7.04 | 8.18 | 8.31 |
| Ta | 3.22 | 3.40 | 3.25 | 2.76 | 2.83 | 4.47 | 2.29 | 2.30 | 3.60 | 3.56 |
| Pb | 2.59 | 2.71 | 2.45 | 2.19 | 2.30 | 3.73 | 1.61 | 1.27 | 2.91 | 2.90 |
| Th | 3.81 | 3.89 | 3.71 | 3.14 | 3.27 | 6.09 | 2.40 | 2.36 | 3.87 | 3.86 |
| U | 1.07 | 1.08 | 0.99 | 0.86 | 0.92 | 1.66 | 0.73 | 0.72 | 0.90 | 0.88 |

## 3. Samples and Analytical Method

Non-nephelinite rock samples, likely representing a shield-building rather than post-shield stage of volcanism [10], were collected from Pohnpei Island at locations shown in Figure 1. The studied basaltic lavas are porphyritic with a gray surface and brown or black patches (Figure 2a–d) containing small phenocrysts (Figure 2e–l). Olivine, clinopyroxene, and plagioclase appear as microphenocrysts in the matrix, with only a few clinopyroxene phenocrysts occurring in samples S7 and S8 (Figure 2e,f,k,l). Fe–Ti oxides, magnetite, and ilmenite occur ubiquitously as inclusions, or are intergrown with olivine and clinopyroxene phenocrysts (Figure 2i–l). Complex textural features such as corroded cores and overgrown rims are common in phenocrysts in all samples.

Whole-rock major-element compositions were determined by X-ray fluorescence spectroscopy (PANalytical Axios$^{MAX}$) at the Key Laboratory of Submarine Geosciences (KLSG), Second Institute of Oceanography, Ministry of Natural Resources, Hangzhou, China, with an analytical precision of better than ±2% to 5%. Trace-element compositions were determined by inductively coupled plasma–mass spectrometry (ICP–MS; Thermo Fisher iCAP RQ equipped with a Cetac ASX-560 AutoSampler)

after complete dissolution, at Guizhou Tongwei Analytical Technology Co., Ltd. (Guizhou, China). US Geological Survey standard W-2a was used as reference standard and crossed-checked with BHVO-2 (Supplementary Table S1). Instrumental drift mass bias was corrected with internal spikes and external monitors. The ICP–MS procedure followed that of Eggins et al. [19], with modifications as described in Kamber et al. [20] and Li et al. [21]. Analysis results are given in Table 1.

Major- and trace-element compositions of phenocrysts of olivine, clinopyroxene, and feldspar were determined using electron probe microanalysis (EPMA; JEOL JXA-8100 Super-probe) at the KLSG, with a 15 kV accelerating voltage, 300 nA (olivine) or 20 nA (clinopyroxene and feldspar) beam current, and 5 μm beam diameters. The Ti counting time was 120 s for olivine, with the procedure for other elements following that of Sobolev et al. [22]. For clinopyroxene and feldspar, the counting times for all major elements followed Li et al. [23]. A range of natural minerals and synthetic oxides of known compositions was used for calibration, with ZAF data correction.

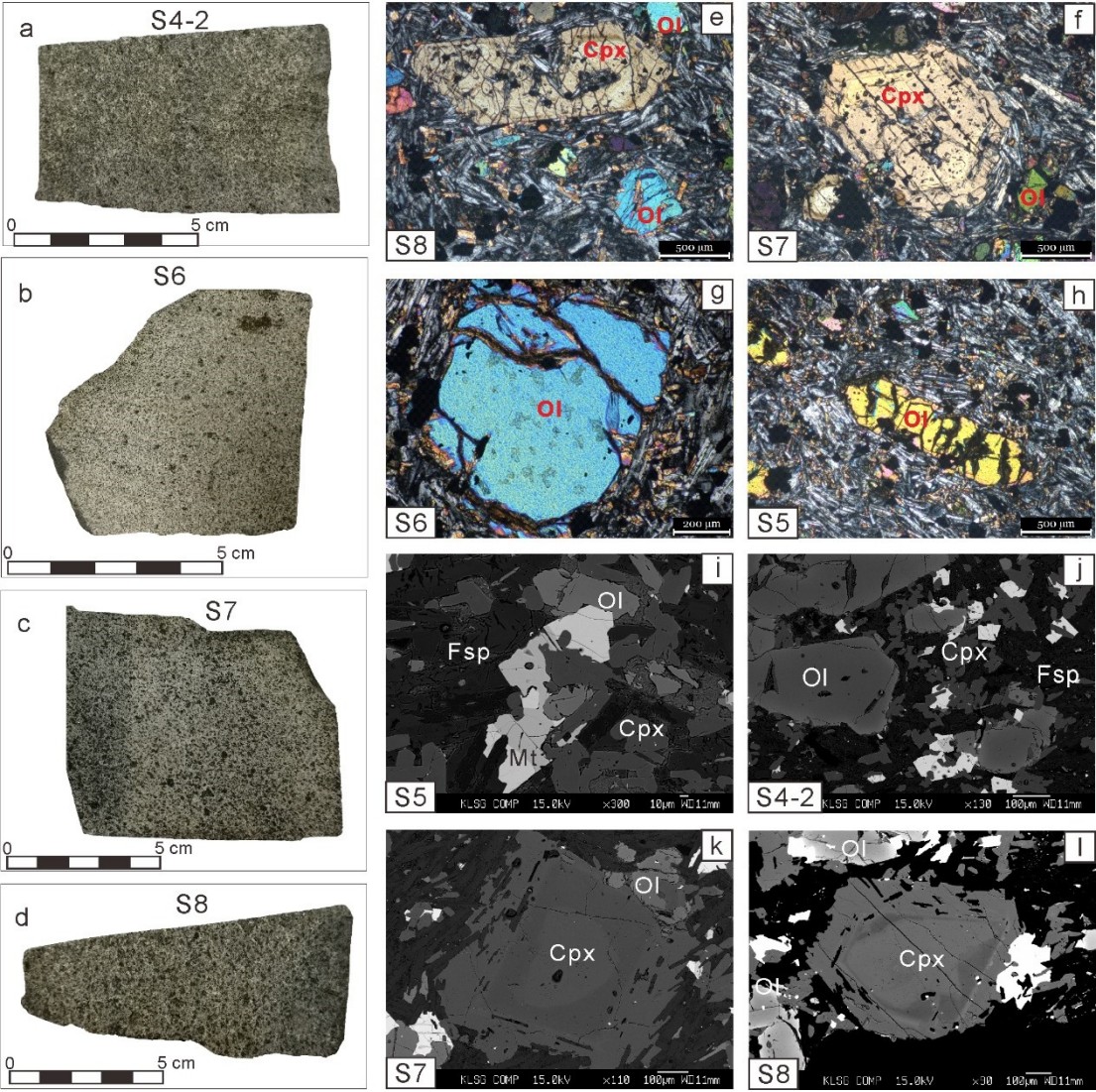

**Figure 2.** Hand specimens (**a**–**d**), photomicrographs showing textural and mineralogical characteristics (**e**–**h**), and back-scattered electron (BSE) images (**i**–**l**) of lavas from Pohnpei Island. Abbreviations: Ol, olivine; Cpx, clinopyroxene; Fsp, feldspar.

## 4. Results

### 4.1. Whole-Rock Major- and Trace-Element Compositions

#### 4.1.1. Major Elements

The studied Pohnpei lavas had undergone slight alteration with loss on ignition (LOI) values of 0.22–2.43 wt.%, and our data for the fresh lavas are within the range of major-element compositions reported previously [9,10,18]. They have $SiO_2$ and MgO contents of 44.78–48.24 and 4.46–11.95 wt.%, respectively, and $Mg^{\#}$ ($100 \times$ mol MgO/(MgO + $FeO^T$)) values of 40–62. The samples have higher total Fe oxide ($FeO^T$) contents and lower $Na_2O$ contents than those of the Carlsberg Ridge MORBs (mid-ocean ridge basalts) [24,25], indicating that they are derived from deeper mantle sources with lower degrees of melting than MORB sources [26,27]. The Pohnpei lavas are enriched in alkali elements ($Na_2O + K_2O$) and are classified as basalt, tephrite basanites, and minor trachybasalts in the ($Na_2O + K_2O$)–$SiO_2$ plot (Figure 3). Together with previously reported data, this suggests that both shield and post-shield volcanism on Pohnpei Island were characterized by alkaline lavas [10], unlike those from the Hawaiian Islands and Ontong Java Plateau, where tholeiitic lavas formed in the shield-building stage and alkalic lavas in the post-shield stage [28–30].

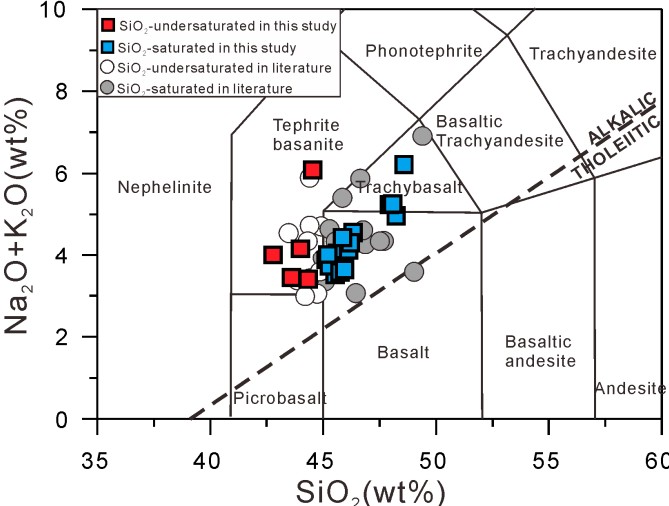

**Figure 3.** ($Na_2O + K_2O$)–$SiO_2$ plot of lavas from Pohnpei Island. Major-element data for lavas from previous studies [9,10,18] are included for comparison.

Pohnpei lavas can be subdivided into $SiO_2$-undersaturated (basanites) and $SiO_2$-saturated (basalt and trachybasalts) lavas, with both likely representing a shield-building rather than post-shield stage of volcanism. The $SiO_2$-undersaturated lavas have significantly higher $TiO_2$ and $FeO^T$ contents than those of $SiO_2$-saturated lavas, with high MgO contents of >8 wt.% (Figure 4). There is no significant difference in $Al_2O_3$, CaO, $Na_2O$, and $K_2O$ contents between the types (Figure 4), indicating that they share similar mantle lithology or mantle source compositions [31]. The CIPW minerals calculation indicates that our basaltic samples comprise olivine (12.9–26.1%), diopside (9.3–29.9%), anorthite (16.0–24.4%), albite (8.2–32.4%), orthoclase (3.3–11.4%), nepheline (0–13.9%), ilmenite (4.5–7.2%), and apatite (1.0–2.3%) (Supplementary Table S5). The $SiO_2$-undersaturated basalts have obviously higher nepheline contents (5.25–13.9% average 8.5%) than $SiO_2$-saturated basalts (0–8.4% average 2.7%).

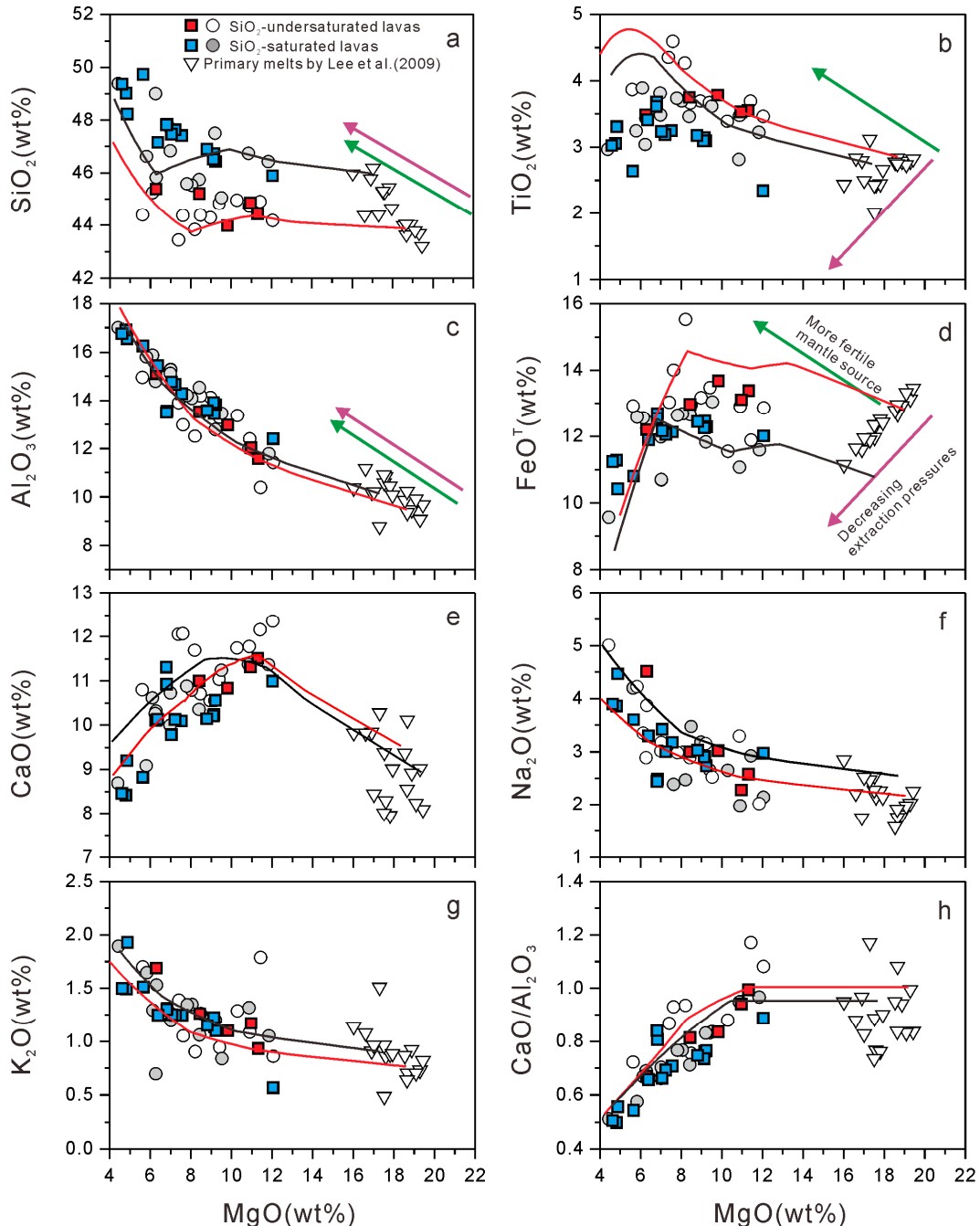

**Figure 4.** Major-element (**a**) $SiO_2$, (**b**) $TiO_2$, (**c**) $Al_2O_3$, (**d**) $FeO^T$, (**e**) CaO, (**f**) $Na_2O$, (**g**) $K_2O$ and (**h**) $CaO/Al_2O_3$ variation diagram as a function of MgO content. Modelled liquid lines of descent for $SiO_2$-saturated (black lines) and $SiO_2$-undersaturated (red lines) lavas, as estimated by the Petrolog3 program [32], are included. Olivine–melt equilibrium models were from Gaetani and Watson [33]; clinopyroxene–melt equilibrium models from Ariskin et al. [34]; and magnetite–melt and ilmenite–melt equilibrium models from Ariskin and Barmina [35] and Nielsen [36], respectively. Major-element data from previous studies are included for comparison [9,10,18]. Compositional trends [37] of melts derived from less to more fertile mantle (green arrows) and from high to low pressures of melt extraction (purple arrows) are shown in (**a**–**d**).

### 4.1.2. Trace Elements

Primitive mantle (PM)-normalized trace-element patterns [38] and chondrite-normalized rare-earth-element (REE) patterns [38] for Pohnpei lavas are shown in Figure 5. The lavas are more

enriched in strongly incompatible elements than global-average enriched MORBs (E-MORBs) [39], with elemental patterns more similar to average OIB patterns [40] (Figure 5). Our new data overlap those of Pohnpei lavas reported by Jackson et al. [10], and plot within the compositional range of lavas from the Caroline Islands (including Chuuk and Kosrae islands). Overall, the $SiO_2$-undersaturated and -saturated lavas display no marked differences in trace-element and REE patterns, with the exception that the former exhibit less Pb depletion (Figure 5a); unlike the Bermuda lavas, whose silica-undersaturated units are more enriched in trace elements [2]. All Pohnpei lavas are enriched in light REEs relative to heavy REEs with $(La/Yb)_N$ ratios of 7–19, and no obvious Eu anomalies (Eu/Eu* = 0.96–1.03, based on chondrite-normalized values) or Sr anomalies (Sr/Sr* = 2 × $Sr_{PM}$/($Pr_{PM}$ + $Nd_{PM}$) = 0.84–1.16, based on primitive mantle-normalized values [41,42])(Figure 5b), indicating insignificant fractional crystallization of feldspar phenocrysts.

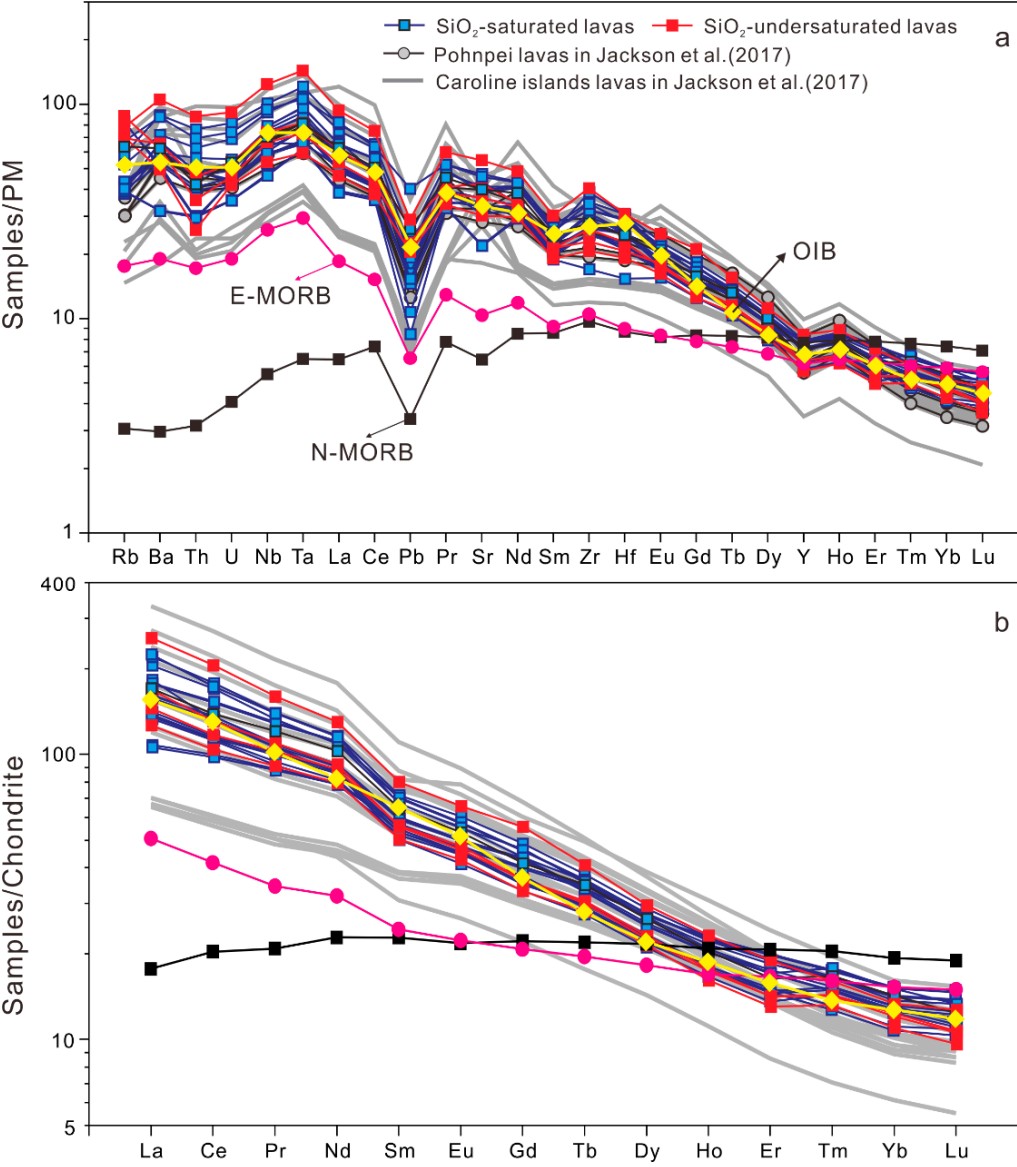

**Figure 5.** (**a**) Primitive mantle (PM [38])-normalized trace-element patterns and (**b**) chondrite-normalized [38] rare-earth-element (REE) patterns of Pohnpei lavas. N-MORB (normal MORB) and E-MORB (enriched MORB) trace-element data are from Gale et al. [39], and ocean island basalts (OIB) data from Sun and McDonough [40]. Data for Caroline Islands lavas (Chuuk, Pohnpei, and Kosrae) are from Jackson et al. [10].

### 4.2. Major and Minor Elements in Minerals

#### 4.2.1. Olivine

EPMA data for olivine crystals are listed in Supplementary Table S2. Olivine displays slight compositional zonation with a darker color (higher Fo; Fo = $100 \times$ mol MgO/(MgO + FeO$^T$) in the large cores and lighter color (lower Fo) in the very thin rims (Figure 2j). These phenocrysts have low Fo values of 66–74, indicating they were crystalized from an evolved mantle-derived melt. Their high CaO contents (0.14–0.40 wt.%; Figure 6a) are also indicative of a magmatic rather than mantle xenolith origin [43,44]. Their Ni and Mn contents are positively and negatively correlated with Fo values, respectively, as commonly observed in OIB and MORB (Figure 6b,c) [22,45]. Fe/Mn ratios vary from 55.5 to 67.8, with most being in the range of olivine crystallized from peridotite-derived melt rather than pyroxenite-derived melt (Figure 6d).

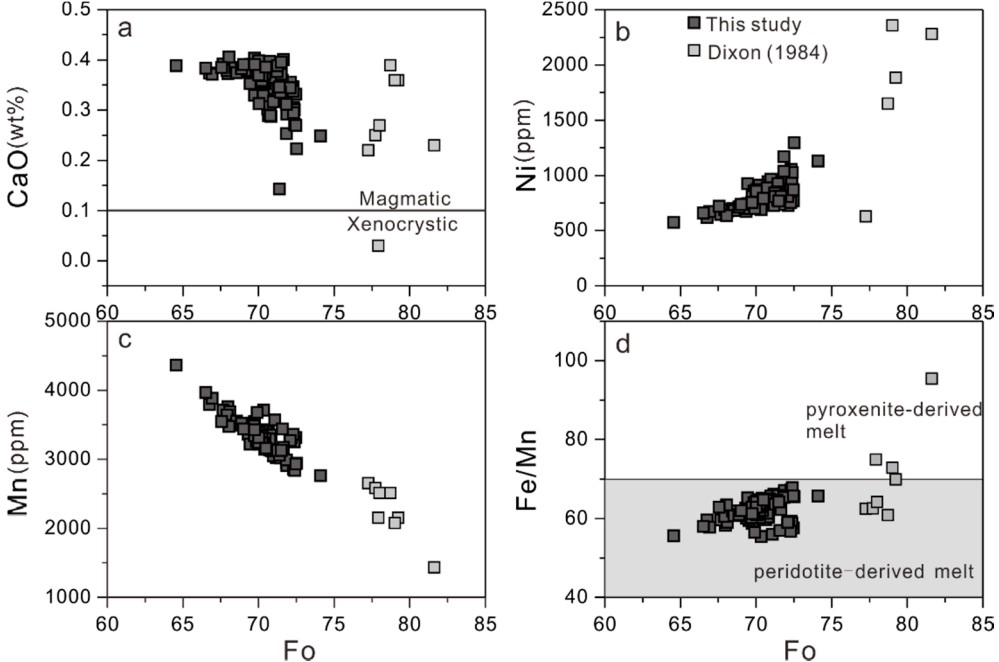

**Figure 6.** Pohnpei basalt olivine CaO (**a**), Ni (**b**), Mn (**c**) contents, and Fe/Mn ratios (**d**) vs. Fo values. Data for Pohnpei olivines from Dixon et al. [9] are plotted for comparison.

#### 4.2.2. Clinopyroxene

EPMA data for clinopyroxenes are listed in Supplementary Table S3. The clinopyroxenes predominantly have compositions of diopside (Di) and minor augite (Au); Wo42-48, En36-45, Fs9-15. Some exhibit oscillatory zoning (Figure 2e,f,k,l). Bright domains in BSE images have lower Mg$^\#$ values and higher TiO$_2$ contents than dark domains (Supplementary Figure S1), and Al$_2$O$_3$, CaO, Na$_2$O, and MnO contents show no systematic variations. The clinopyroxenes have MgO contents of 11.12–16.00 wt.% with Mg$^\#$ values of 67–82, suggesting variable degrees of fractional crystallization. Their FeO$^T$ contents increase sharply with decreasing MgO content between Mg$^\#$ values of 82 and 75, but increase more gently as Mg# values decrease below 75 (Supplementary Figure S2), suggesting that Fe–Ti oxides might have begun to crystallize at Mg#$_{Cpx}$ = ~75 [46].

#### 4.2.3. Feldspar

Feldspars occur mainly in the groundmass, usually as intergrowths with fine-grained clinopyroxene (Figure 2). They are chemically homogeneous with no compositional zonation (Figure 2). Feldspar major-element compositions are listed in Supplementary Table S4. They have wide range of

An values (12–62), and can be classified as plagioclase (labradorite and andesine) and alkali feldspar (anorthoclase and sanidine; Supplementary Figure S3). The strong correlation of CaO, $Al_2O_3$, and $Na_2O$ contents with $SiO_2$ content (Supplementary Figure S3) suggests a steady evolutionary state in the late stage of magma differentiation.

## 5. Discussion

### 5.1. Rock Alteration

The low Pohnpei lava LOI values (0.22–2.43 wt.%; Table 1) and positive correlation of fluid-mobile (e.g., Ba and U; Figure 7a,b) and -immobile (e.g., Th and Nb) elements with Zr (Figure 7c,d) indicate weak or insignificant alteration effects, consistent with petrographic observations of fresh cores and a lack of mineral alteration.

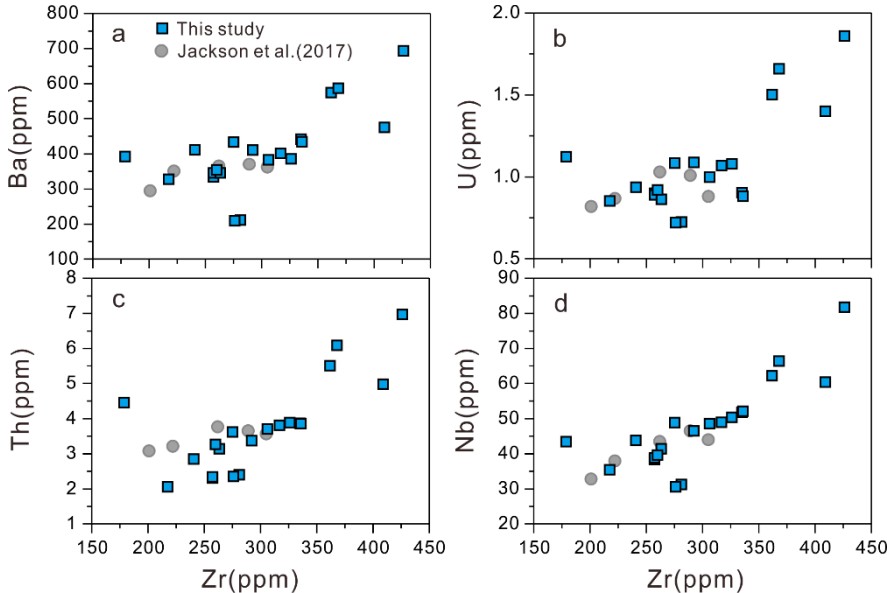

**Figure 7.** Variations of (**a**) Ba, (**b**) U, (**c**) Th, and (**d**) Nb contents with Zr contents of Pohnpei lavas. Previously reported data (light-gray circles) [10] are also plotted.

### 5.2. Primary Magma Composition and Mantle Melting Process

Very primitive magma that has crystallized only olivine is often used as a starting composition when modelling magma compositions. For example, terrestrial magmas and Hawaiian basalts with MgO contents of >8.5 and >9 wt.%, respectively, were used in modelling by Lee et al. [47] to determine parental magma compositions. The advantage of that model is that it is an $SiO_2$-based barometer that is less sensitive to variations in mantle composition than those based on incompatible trace elements [47]. For the Pohnpei lavas, the modelled liquid lines of descent (LLDs) indicate that those with MgO contents of >9 wt.% might have predominantly crystallized olivine, with simultaneous crystallization of minor clinopyroxene having little effect on melt compositions (Figure 4). Samples with MgO contents of >9wt.% were therefore chosen as starting compositions here to determine primary magma compositions using the model of Lee et al. [47]. Calculations continued until the melt was in equilibrium with $Fo_{90}$ olivine. Modelling was based on 16 samples, and previously published data were also used. Results indicate that 17–31% olivine must be added back to reach $Fo_{90}$ equilibrium in Pohnpei Island basalts.

The calculated compositions of primary melts are shown in Table 2 and Figure 4. The modelled primary magma exhibits moderate chemical variations, with those of $SiO_2$ and $Al_2O_3$ with MgO in Pohnpei primary magmas being affected by the mantle source and/or melt-extraction pressure (Figure 4a,c; [37]). A fertile mantle source has higher $SiO_2$, $Al_2O_3$, $FeO^T$, and $TiO_2$ contents and lower

MgO content than less-fertile mantle [48,49]. Melt $SiO_2$ and $Al_2O_3$ contents increase and FeO and MgO contents decrease with decreasing melting pressure [37]. Previous studies have suggested that the mantle source of Pohnpei Island is relatively homogenous [10,13], and the variations of $FeO^T$ and $TiO_2$ with MgO (Figure 4b,d), and $SiO_2$ and $Al_2O_3$ with $FeO^T$ (not shown) thus reflect the effect of melt extraction pressures rather than mantle composition.

**Table 2.** Estimated primary melt compositions and mantle melting conditions for the Pohnpei lavas.

| Samples | $SiO_2$ | $TiO_2$ | $Al_2O_3$ | $FeO^T$ | MnO | MgO | CaO | $Na_2O$ | $K_2O$ | T (°C) | P (GPa) |
|---|---|---|---|---|---|---|---|---|---|---|---|
| WF-2 | 45.26 | 2.00 | 10.57 | 11.97 | 0.15 | 17.55 | 9.37 | 2.54 | 0.48 | 1537 | 3.4 |
| NAN-1 | 43.68 | 2.79 | 9.06 | 13.14 | 0.14 | 19.33 | 9.01 | 2.02 | 0.73 | 1616 | 4.7 |
| WF-1 | 44.03 | 2.79 | 9.46 | 12.93 | 0.14 | 18.91 | 8.91 | 1.79 | 0.92 | 1593 | 4.3 |
| NAN-2 | 43.19 | 2.82 | 9.67 | 13.46 | 0.13 | 19.47 | 8.08 | 2.25 | 0.82 | 1626 | 5.1 |
| S7 | 45.26 | 2.43 | 10.82 | 12.36 | 0.13 | 17.62 | 8.27 | 2.14 | 0.86 | 1533 | 3.3 |
| P-02 | 45.44 | 2.44 | 10.43 | 12.52 | 0.14 | 17.85 | 7.94 | 2.26 | 0.88 | 1538 | 3.4 |
| S8 | 45.31 | 2.43 | 10.89 | 12.35 | 0.14 | 17.55 | 8.00 | 2.28 | 0.96 | 1530 | 3.3 |
| PON13-26 [a] | 43.62 | 2.84 | 9.33 | 12.68 | 0.14 | 18.70 | 10.11 | 1.75 | 0.70 | 1597 | 4.4 |
| PON13-06 [a] | 43.97 | 2.77 | 9.85 | 12.78 | 0.15 | 18.57 | 9.36 | 1.57 | 0.86 | 1583 | 4.1 |
| PON13-13 [a] | 43.78 | 2.73 | 9.88 | 13.33 | 0.15 | 19.13 | 8.21 | 1.97 | 0.70 | 1602 | 4.5 |
| P-2 [b] | 45.77 | 2.79 | 10.18 | 11.60 | 0.13 | 16.93 | 9.83 | 1.74 | 0.91 | 1509 | 2.9 |
| P18 [b] | 44.39 | 3.12 | 8.76 | 11.88 | 0.15 | 17.32 | 10.28 | 2.47 | 1.51 | 1549 | 4.0 |
| P23 [b] | 45.97 | 2.43 | 10.36 | 11.14 | 0.14 | 16.05 | 9.83 | 2.85 | 1.14 | 1486 | 3.0 |
| P-3B [c] | 44.36 | 2.83 | 11.17 | 11.64 | 0.15 | 16.63 | 9.82 | 2.21 | 1.08 | 1518 | 3.4 |
| P-9 [c] | 44.04 | 2.75 | 10.22 | 12.97 | 0.13 | 18.69 | 8.54 | 1.91 | 0.64 | 1584 | 4.1 |
| P27 [c] | 46.17 | 2.49 | 10.20 | 11.96 | 0.13 | 17.04 | 8.43 | 2.52 | 0.96 | 1508 | 3.0 |
| Average | 44.64 | 2.65 | 10.05 | 12.42 | 0.14 | 17.96 | 9.00 | 2.14 | 0.88 | 1557 | 3.8 |

Note: Major element contents are in wt%; [a], data from Jackson et al. [10]; [b], data from Dixon et al. [9]; [c], data from Mattey [18].

Fitton et al. [50] suggested that varying degrees of partial melting of a uniform peridotite source produce an array approximately parallel to the Icelandic array in a (Zr/Y)–(Nb/Y) diagram. Our Zr/Y and Nb/Y data cover a narrow range and define an array parallel to that of Icelandic lavas, and within that for the Hawaiian–Emperor Chain [51] (Supplementary Figure S4). This further supports the variability of basalt composition being controlled by the degree of melting of a compositionally homogenous mantle source. Mantle upwelling is accompanied by continued decompression melting, and primary magma formed at higher pressures thus has lower $SiO_2$ and $Al_2O_3$ contents, but higher $FeO^T$, MgO, $TiO_2$, and $P_2O_5$ contents, and vice versa [52,53]. The notable correlation of primary magma composition with mantle melting pressure thus further indicates that compositional variations in Pohnpei primary magmas can best be explained by melt extraction in the melting column at varying mantle depths/pressures.

The pressure (P) and temperature (T) of mantle melting were calculated by the method of Lee et al. [47] using estimated primary-melt compositions, with results listed in Table 2. The estimated melting pressures of 2.9–5.1 GPa (average 3.8 ± 0.7 GPa; 1σ) with melting temperatures are 1486–1626 °C (average 1557 ± 43 °C; 1σ; Table 2). The estimated P–T melting conditions for Chuuk and Kosrae island mantle are 4.1 GPa and 1592 °C, and 4.5 GPa and 1590 °C, respectively (excluding anomalous data for samples KOS13-19 and KOS20). These values are obviously higher than those of the melting conditions of MORB sources (P = 0.7–1.7 GPa; T = 1300–1400 °C; corresponding to 30–50 km depth) and primary magmas from the Hawaiian Island and Louisville seamount chains (Figure 8a). The Caroline Islands mantle underwent 0–15% melting (Figure 8a), as estimated by the model of Lee et al. [47], consistent with estimates based on trace-element compositions (Figure 8b). This degree of melting is much lower than that of the Hawaiian Islands (10–25%; Figure 8a), consistent with the contrasting occurrence of alkaline and tholeiitic lavas in shield-building volcanism. This also indicates that the $SiO_2$-undersaturated lavas were generated at higher mantle temperatures and pressures than $SiO_2$-saturated lavas (Figure 8a). Higher mantle temperatures caused melting to begin at greater pressure (depth) [54], explaining the higher $TiO_2$ and $FeO^T$ contents of $SiO_2$-undersaturated samples than $SiO_2$-saturated samples.

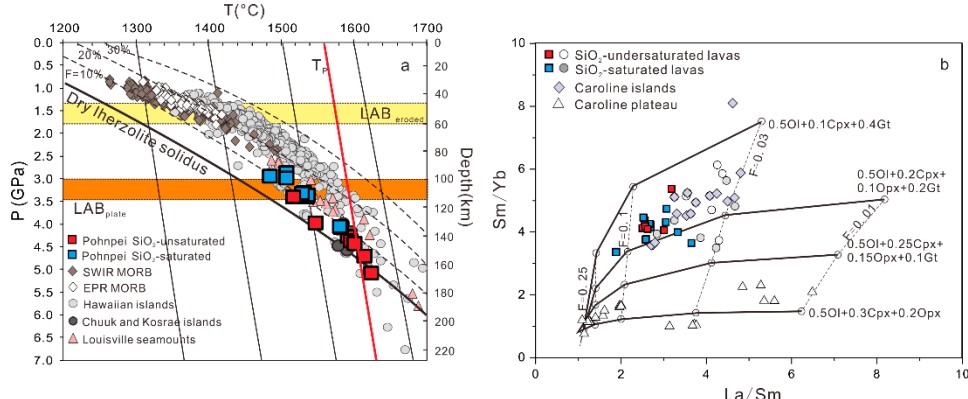

**Figure 8.** (**a**) Temperatures and pressures calculated for Pohnpei lava, based on the model of Lee et al. [47]. Lherzolite solidus and melt fraction isopleths are from Katz et al. [55]. Curved lines represent adiabatic melting conditions; near-vertical lines (gray and red) represent solid-mantle adiabatic conditions. Hawaiian Island basalt data are from Lee et al. [47]; Southwest Indian Ocean Ridge (SWIR) and East Pacific Rise (EPR) MORB data from Gale et al. [39]; Chuuk and Kosrae Island basalt data from Jackson et al. [10]; and Louisville Seamount basalt data from Beier et al. [56] and Vanderkluysen et al. [51]. Basalts with MgO contents of >9 wt.% were corrected to olivine Fo90 following the method of Lee et al. [47]. (**b**) (Sm/Yb)–(La/Sm) plot based on the model of Zhang et al. [13] and references therein. Abbreviations: Ol, olivine; Cpx, clinopyroxene; Opx, orthopyroxene; Gt, garnet.

In the (Sm/Yb)–(La/Sm) diagram (Figure 8b), most Pohnpei lavas plot in the field between 0.5Ol + 0.1Cpx + 0.4Gt and 0.5Ol + 0.2Cpx + 0.1Opx + 0.2Gt, suggesting garnet rather than spinel as a residual phase in the mantle source during melting. In addition, the estimated melting pressures for Pohnpei lavas form an array from 2.5 GPa (75–80 km), intersecting the dry lherzolite solidus at ~5 GPa (150–180 km), corresponding to a potential mantle temperature of 1560–1600 °C, ~200 °C above the average MORB mantle and very similar to those inferred for Hawaiian primary magmas [47]. The mantle melting process beneath Pohnpei Island thus took place mainly in a garnet–peridotite field.

### 5.3. Modeling of Liquid Lines of Descent and Mineral Crystallization Sequence

Magma LLDs provide a wealth of information on mineral assemblages, and primary magma composition and its physico-chemical conditions including T, P, $H_2O$ content, and oxygen fugacity ($fO_2$) [57–61]. Here we used the Petrolog3 software package [32] to simulate the magma differentiation pathway.

Primary magma compositions restored by the method of Lee et al. [47] were set as the starting composition, and an olivine–clinopyroxene–magnetite–ilmenite assemblage as liquidus minerals for forward modeling. The distribution of estimated primary magma compositions (Figure 4) was produced by varying the extent of melting within the mantle column. We modeled two differentiation paths to fit the geochemical data: one involving a primary magma composition of NAN-1, and the other a primary magma composition of P-2 (Table 2). Variable pressure, $H_2O$ content, and $fO_2$ values were assigned as starting physicochemical conditions to reconcile modelled differentiation pathways with geochemical trends [62].

Oxygen fugacity determines the $Fe^{3+}/\Sigma Fe$ composition of the initial melt and controls the crystallization of Fe–Ti oxides in magma systems [61]. The $fO_2$ values of QFM + 1.5 and QFM + 1 were set for $SiO_2$-undersaturated and -saturated primary magma, respectively, by fitting the timing of magnetite crystallization. The major effect of $H_2O$ content is to suppress the crystallization of plagioclase [58], which itself is not a major mineral phase. The $H_2O$ content is therefore not an important factor in our model and was set arbitrarily at 0.2 wt.%. The crystallization pressure was set at 3 kbar assuming the crustal thickness of Pohnpei Island was 9 km.

Modelled LLDs generally match observed evolutionary trends well for major elements in both $SiO_2$-undersaturated and -saturated lavas (Figure 4), with both lava series displaying parallel evolutionary trends during olivine crystallization. However, the abrupt increase in $SiO_2$, $Al_2O_3$, $Na_2O$, and $K_2O$ contents (Figure 4a,c,f,g), and the decrease in $FeO^T$ and CaO contents and $CaO/Al_2O_3$ ratios at higher MgO values (Figure 4d,e,h), indicate earlier clinopyroxene and magnetite crystallization in $SiO_2$-undersaturated lavas than the saturated lavas. The abrupt $TiO_2$ decrease at higher MgO contents reflects late crystallization of ilmenite in the model. The increase of $Al_2O_3$, $Na_2O$, and $K_2O$ contents with decreasing MgO content (Figure 4c,f,g), and the lack of an Eu anomaly (Figure 5) suggest that plagioclase is not a major crystallizing phase, which is consistent with the petrological observations (Figure 2e-l).

In summary, Pohnpei lavas have undergone sequential crystallization of olivine, clinopyroxene, Fe–Ti oxides (magnetite and ilmenite), and minor feldspar. Mineral assemblages inferred from major- and trace-element compositions are consistent with those observed in microscopic and BSE images (Figure 2e–l).

### 5.4. Lithospheric Thickness Control of Mantle Melting and Magma Evolution

Previous studies have indicated that the age and thickness of the oceanic and continental lithosphere exert an important influence on the compositional variation of intraplate OIB-type lavas [17,37,52,53,63,64]. For example, Niu et al. [53] demonstrated that, on a global scale, the geochemical compositions of OIBs erupted on thick lithosphere are consistent with a low degree of partial melting under high pressure, while those erupted on thinner lithosphere exhibit a higher degree of partial melting, with melting ceasing at shallower depths. Sun et al. [37] found that lithospheric thickness variations exert primary control on the geochemistry of intra-continental basalts. Zhang et al. [13] have shown that Caroline Plateau alkaline lavas have higher $SiO_2$ and $Al_2O_3$ contents and lower $FeO^T$ contents and Sm/Yb ratios than those of the Caroline Islands. Our new data for Caroline Island lavas also indicate much higher $(La/Yb)_N$ and $(Sm/Yb)_N$ ratios than those of the Caroline Plateau. Taken together, these data suggest that the Caroline Islands were formed by lower degrees of melting at greater depths, due to a thicker overlying lithosphere than that of the Caroline Plateau at the time of volcanism [13].

The Louisville Seamount Chain on the Pacific Plate preserves continuous, predominantly age-progressive (>70 Ma) volcanism [56,65], with a magma composition considered to be controlled by the varying lithospheric thickness [37,56]. Although the Caroline Islands (including Chuuk, Pohnpei, and Kosrae) also display an age-progressive trend, the old (ca. 160 Ma) lithospheric thickness has been relatively constant at ~90 km (based on the model of Humphreys and Niu [52]). The new data for $(La/Yb)_N$ and $(Sm/Yb)_N$ ratios of basaltic lavas from both Pohnpei Island and the Louisville Seamount Chain indicate a statistically positive correlation with lithospheric thickness (Figure 9), supporting first-order control of lithospheric thickness on the chemical composition of ocean-island lavas, although a role of mantle source heterogeneity is not precluded [17,37,52,53,63,64].

The lithospheric mantle can also affect magma evolution from source to eruption, through melt–lithosphere interaction and fractional crystallization in the crust [66]. Lithospheric mantle has been suggested to act as a chemical filter, modifying the composition of asthenosphere-derived magma to a degree depending on lithospheric thickness [4,6]. Metasomatized lithosphere is also thought to contribute to the compositional diversity of OIBs [5,67]. We therefore speculate that the composition of Pohnpei lavas was, to some extent, modified when penetrating the thick lithosphere, although this effect was not evaluated here. On the other hand, lithospheric thickness can indirectly affect the crystallization sequence of OIBs by determining their primary composition. For example, the early onset of magnetite crystallization (MgO > 8 wt.%) in Pohnpei lavas would have resulted from initially high $FeO^T$ content and oxygen fugacity, while the late crystallization of feldspar would have been due to an initially low $Al_2O_3$ content, typical of a low degree of melting in deep mantle overlying thick lithosphere [52,53].

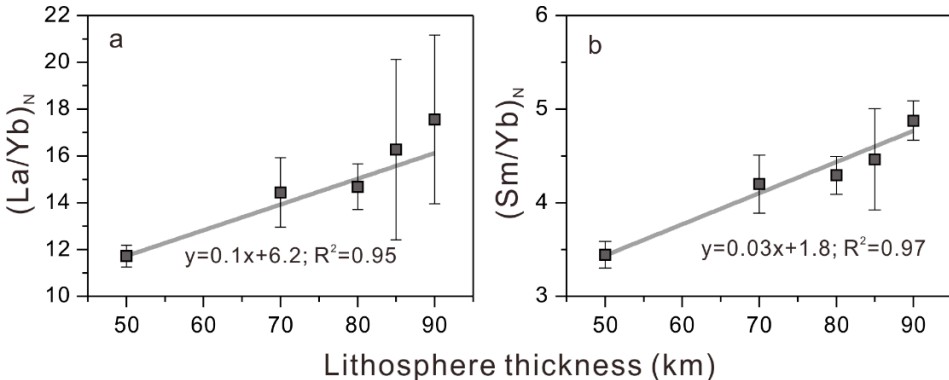

**Figure 9.** Variations of (**a**) $(La/Yb)_N$ and (**b**) $(Sm/Yb)_N$ ratios with lithospheric thickness beneath Pohnpei Island and Louisville seamounts. The ratios were averaged at 5 km intervals of lithospheric thickness for basalts with MgO contents of >6 wt.% for Louisville seamounts and >9 wt.% for Pohnpei Island. Error bars represent ±1SD. Louisville seamount data are from Beier et al. [56] and Vanderkluysen et al. [51].

## 6. Conclusions

(1)   Pohnpei Island lavas are all alkalic and can be classified as $SiO_2$-undersaturated or $SiO_2$-saturated, with the former having higher $TiO_2$ and $FeO^T$ contents but with no distinct trace-element composition, suggesting they formed by partial melting of a compositionally homogenous mantle source at varying depth.

(2)   The primary magma underwent sequential crystallization of olivine, clinopyroxene, and Fe–Ti oxides (magnetite and ilmenite), as well as a minor plagioclase and alkali feldspar. Early magnetite crystallization (>8 wt.% MgO) and late feldspar crystallization were due to initially high $FeO^T$ and low $Al_2O_3$ contents in the primary magma, which resulted from a low degree of melting in response to a thick lithosphere (the "lid effect").

(3)   The Pohnpei Island lavas formed at mantle-melting pressures of 2.9–5.1 GPa (average 3.8 ± 0.7 GPa) and temperatures of 1486–1626 °C (average 1557 ± 43 °C) in a garnet stability field. Our trace-element data for basalt lavas of Pohnpei Island and Louisville seamounts indicate a strong correlation between $(La/Yb)_N$ and $(Sm/Yb)_N$ ratios and lithospheric thickness, supporting first-order control of lithospheric thickness on chemical compositions of ocean-island lavas.

**Supplementary Materials:** The following are available online at http://www.mdpi.com/2075-163X/10/9/816/s1, Figure S1: Compositional zoning in BSE images (a, d) and chemical compositions of mineral zones for clinopyroxenes (b, c, and e, f). Figure S2: Variation of represent major elements versus Mg# values for clinopyroxenes in the Pohnpei lavas. Figure S3: (a) Feldspar compositional range projected into the Or-Ab-An ternary and (b) relationships between represent major elements ($Al_2O_3$, $Na_2O$, and $CaO$) contents and $SiO_2$ contents in feldspars in lavas from Pohnpei island. Figure S4: Variation of Pohnpei samples in Nb/Y and Zr/Y. Iceland array is from Fitton et al. [50]. Field for the Hawaiian-Emperor chain is from Vanderkluysen et al. [51]. Table S1: Major and trace element compositions of standard materials BHVO-2 and W-2a. Table S2: Major and minor element compositions of olivine phenocrysts in Pohnpei island lavas. Table S3: Major and minor element compositions of clinopyroxene phenocrysts in Pohnpei Island lavas. Table S4: Major and minor element compositions of Feldspar phenocrysts in Pohnpei island lavas. Table S5: CIPW minerals calculated for lavas from Pohnpei Island.

**Author Contributions:** Conceptualization, Z.-G.L. and T.Z.; methodology, T.Z. and Z.-G.L.; software, T.Z.; validation, T.Z., Z.-G.L., Y.-H.D., and X.-P.L.; formal analysis, J.-H.Z., L.C., and J.-Q.L.; writing (original draft preparation), T.Z.; writing (review and editing), T.Z. and Z.-G.L.; and funding acquisition, Z.-G.L., Y.-H.D., and X.-P.L. All authors have read and agreed to the published version of the manuscript. Please turn to CRediT taxonomy for term explanations. Authorship is limited to those who have contributed substantially to the work.

**Funding:** This research was funded by Resources and Environment Projects of China Ocean Mineral R&D Association (DY135-E2-2-01, DY135-G2-1-03), the Second Institute of Oceanography, the MNR of China (JT1301, JT1504), Key project of the National Natural Science Foundation of China (41230960), and SDUST Research Fund (2015TDJH101).

**Acknowledgments:** We thank the anonymous reviewers whose comments improved the manuscript.

**Conflicts of Interest:** The authors declare no conflict of interest.

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
