# Peer review of "Geochemical Constraints on Mantle Melting and Magma Genesis at Pohnpei Island, Micronesia"

_minerals, doi:10.3390/min10090816_

Round 1
Reviewer 1 Report
The manuscript "minerals-867382: Geochemical constraint on mantle melting and magmagenesis of Pohnpei island in the Micronesia, West Pacific by Song and others" reports geochemical results (whole-rock major and trace elements and major mineral compositions) of basaltic rocks from the Pohnpei island of Micronesia. Based on their geochemical data and mantle-model calculations, the above authors interpret lower degree mantle melting at greater depths (> 90 km) and ascribe it to thick lithosphere.
The geological background, data presentation and interpretation is nicely done, hence I recommend the ms for publication in journal "Minerals". However, there are several minor (but important issues) that need to be further improved before the manuscript should be acceptable fro publication.
I have marked most of my comments on the attached annotated pdf file and hope the authors will find the useful for the improvement of their ms.
A few main points: Sample location is not given in the map, so it is really important to show them. As discussed by the authors, previous studies reported main lava series and later stage volcanism in the island. This is not well treated in the ms. In its current situation, it is not clear which series rocks have been reported in this ms. Then, based on small number of samples, authors have constructed their model of deeper lithospheric mantle plus the origin of olivine as formed from an evolved mantle-derived melt. This could be true but I wonder if all the rock types in Pohnpei have formed from the same source and under such higher P-T conditions (1400 ~ 1600 T and 2.5 ~ 5 GPa).
For other specific comments, please see the attached pdf file.
Although, not a major issue, but I recommend authors to carefully check and correct the language wherever needed (although, I have marked some corrections).
With best regards
Reviewer 1

Author Response
Dear Editor,
Thank you very much for handling our manuscript. Thanks also to the two anonymous referees for their constructive review comments, which are helpful for us to better improve our manuscript. All the comments have been carefully considered, and corresponding modifications have been made wherever necessary. Our responses (colored and in italic) to the editor’s and reviewers’ comments (black and plain style) now follow.
To Reviewer 1:
The manuscript "minerals-867382: Geochemical constraint on mantle melting and magmagenesis of Pohnpei island in the Micronesia, West Pacific by Song and others" reports geochemical results (whole-rock major and trace elements and major mineral compositions) of basaltic rocks from the Pohnpei island of Micronesia. Based on their geochemical data and mantle-model calculations, the above authors interpret lower degree mantle melting at greater depths (> 90 km) and ascribe it to thick lithosphere.
The geological background, data presentation and interpretation is nicely done, hence I recommend the ms for publication in journal "Minerals". However, there are several minor (but important issues) that need to be further improved before the manuscript should be acceptable fro publication.
I have marked most of my comments on the attached annotated pdf file and hope the authors will find the useful for the improvement of their ms.
A few main points: Sample location is not given in the map, so it is really important to show them. As discussed by the authors, previous studies reported main lava series and later stage volcanism in the island. This is not well treated in the ms. In its current situation, it is not clear which series rocks have been reported in this ms. Then, based on small number of samples, authors have constructed their model of deeper lithospheric mantle plus the origin of olivine as formed from an evolved mantle-derived melt. This could be true but I wonder if all the rock types in Pohnpei have formed from the same source and under such higher P-T conditions (1400 ~ 1600 T and 2.5 ~ 5 GPa).
Response: Thanks. The sample location is given in a new map (see Figure 1). Previous studies reported main lava series and later stage volcanism in the island. But they are only used as background introduction, not the focus of this paper. The rocks are not nephelinites and thus likely formed in a shield-building stage rather than in a post-shield stage of volcanism, although the age data was not analyzed for them (see section 3 “Samples and analytical method”).
This paper collected almost all the published data of Pohnpei Island lavas so the results reflect the whole picture of their mantle source and melting condition. Since these lavas experienced variable extent of evolution, we only chosen those just crystallize olivine. Calculation results suggested that these samples were formed under higher P-T conditions. In addition, Pohnpei Island have thicker lithospheric mantle than Hawaii Islands, and the result accords with the conclusion that thicker overlying lithosphere can result in melting at greater depths and temperature.
For other specific comments, please see the attached pdf file.
Response: Thanks for the improvement for our manuscript. We revised and improved the manuscript following the suggestions.
Although, not a major issue, but I recommend authors to carefully check and correct the language wherever needed (although, I have marked some corrections).
Response: Thanks. This has been done and marked in reviewed manuscript. Some specific comments are answered in the attached pdf file of the first manuscript.
With best regards

Reviewer 2 Report
Review of Zong et al., “Geochemical constraint on mantle melting and magma genesis of Pohnpei island in the Micronesia, West Pacific.”
This paper provides a new dataset of bulk major and trace element data and in situ olivine, clinopyroxene, and plagioclase compositions for basalts from Pohnpei. The manuscript uses the data to infer the conditions of partial melting and to calculate liquid lines of descent for the samples, and concludes there is no significant difference between the melting conditions of the Si-saturated and Si-undersaturated lavas. Finally, the manuscript attempts to relate broader trends in major and trace element data to regional lithospheric thickness and distance along the hotspot chain.
While the data appear to be of good quality, I generally find that this manuscript overemphasizes discussion that is not well supported by the data, and also underutilizes the available data to more robustly test computerized models like Petrolog. In particular, I find that the paper requires correction of the following major points:
- I have strong doubts that the trends the authors identify in Figure 8 truly exist. The manuscript describes the trends as “statistically positive correlation[s]” but makes no reference to what statistics were used to come to that conclusion. If the correlations are real, it would seem as if the authors could simply plot the trends and associated error envelopes instead of simply placing arrows in the background, which absent a statistical consideration just seem misleading to me. What the assembled data in Figure 8 do show to me is that there is a lot of data for areas overlying relatively thick lithosphere, but very little data for areas with thinner lithosphere. The new data in this paper effectively cover the whole range of data from thinner lithospheric thicknesses, which would seem to preclude the existence of any correlation. Further, if differences in lithospheric thickness is primarily responsible for differences in the extent of partial melting, then it would seem that the better ratio to choose here would be La/Yb, rather than Sm/Yb and Dy/Yb. Similarly, when the authors assess differences in magma productivity between different islands, I am not at all convinced by the trend for Nb/Zr, where gain the data for Pohnpei range to Nb/Zr values both higher and lower than literature data for Chuuk or Kosrae. If the authors want the trends they observe to be considered seriously, they should offer some rigorous statistical consideration, including accounting for differences in data availability, to back up their claims. Alternatively, the authors could also calculate the parental magma compositions for logical groupings of the samples in the figures and compare those to each other.
- The manuscript relies heavily on major element modeling undertaken with Petrolog, however there is little attempt to use the trace element and in situ data (the latter is not considered at all in the Discussion) to add a check on the modeling. For example, the authors could use REE compositions to calculate the extent of melting in a garnet-bearing source and compare this to the results shown in Figure 7. Or, the authors could compare their in situ data to the mineral composition outputs of Petrolog to verify that their selected initial conditions are producing mineral compositions similar to what they actually measure (c.f., Herzberg et al., 2014 EPSL, although in this paper Petrolog is not used in favor of the authors’ own calculations). Calculated LLD could also consider the effects of crustal and lithospheric assimilation on magma compositions; such assimilation has been shown to exert strong controls on trace element compositions in other hotspots (c.f., Class & Goldstein, 1997 EPSL). If the authors want to further try to connect their data to lithospheric thickness, they could test whether there are differences in “canonical” trace element ratios between areas with different lithospheric thicknesses to see if different depths of melting induce differences in the relative abundance of enriched/fusible mantle lithologies.
In summary, I’m left with the feeling that that paper underutilized the reported data, and instead tried to focus on correlations between geochemical and geophysical data that are generally difficult to definitively establish. I don’t think the authors’ ideas are bad ones, just not well supported by the available data. If the authors want to keep the direction of the paper, they should be clearer about the limitations of their data and how they support their hypotheses.
I’ve also left a number of detailed comments in the manuscript PDF for the authors to review.
Reviewer 2, July 2020
Author Response
To Reviewer 2:
Thank you very much for handling our manuscript. Thanks also to the two anonymous referees for their constructive review comments, which are helpful for us to better improve our manuscript. All the comments have been carefully considered, and corresponding modifications have been made wherever necessary. Our responses (colored and in italic) to the editor’s and reviewers’ comments (black and plain style) now follow.
Review of Zong et al., “Geochemical constraint on mantle melting and magma genesis of Pohnpei island in the Micronesia, West Pacific.”
This paper provides a new dataset of bulk major and trace element data and in situ olivine, clinopyroxene, and plagioclase compositions for basalts from Pohnpei. The manuscript uses the data to infer the conditions of partial melting and to calculate liquid lines of descent for the samples, and concludes there is no significant difference between the melting conditions of the Si-saturated and Si-undersaturated lavas. Finally, the manuscript attempts to relate broader trends in major and trace element data to regional lithospheric thickness and distance along the hotspot chain.
While the data appear to be of good quality, I generally find that this manuscript overemphasizes discussion that is not well supported by the data, and also underutilizes the available data to more robustly test computerized models like Petrolog. In particular, I find that the paper requires correction of the following major points:
- I have strong doubts that the trends the authors identify in Figure 8 truly exist. The manuscript describes the trends as “statistically positive correlation[s]” but makes no reference to what statistics were used to come to that conclusion. If the correlations are real, it would seem as if the authors could simply plot the trends and associated error envelopes instead of simply placing arrows in the background, which absent a statistical consideration just seem misleading to me. What the assembled data in Figure 8 do show to me is that there is a lot of data for areas overlying relatively thick lithosphere, but very little data for areas with thinner lithosphere. The new data in this paper effectively cover the whole range of data from thinner lithospheric thicknesses, which would seem to preclude the existence of any correlation. Further, if differences in lithospheric thickness is primarily responsible for differences in the extent of partial melting, then it would seem that the better ratio to choose here would be La/Yb, rather than Sm/Yb and Dy/Yb. Similarly, when the authors assess differences in magma productivity between different islands, I am not at all convinced by the trend for Nb/Zr, where gain the data for Pohnpei range to Nb/Zr values both higher and lower than literature data for Chuuk or Kosrae. If the authors want the trends they observe to be considered seriously, they should offer some rigorous statistical consideration, including accounting for differences in data availability, to back up their claims. Alternatively, the authors could also calculate the parental magma compositions for logical groupings of the samples in the figures and compare those to each other.
Response: Thanks. This is a very helpful suggestion. It has been widely reported that the thickness of the oceanic and continental lithosphere exerts an important influence on the compositional variation of intraplate OIB-type lavas. For example, Niu et al. [48] have demonstrated that on a global scale the geochemical compositions of OIBs erupted on thick lithosphere is consistent with a low extent of partial melting under high pressure, whereas those erupted on thinner lithosphere exhibit a higher extent of partial melting as the cessation of melting occurring at lower depths. In addition, Sun et al. [42] have further showed that lithosphere thickness variation also exerts the primary control on the geochemistry of continental intraplate basalts.
We deal with our new data and the published data from the Louisville Seamount Chains, which have variable lithosphere thickness. We average the trace element ratios (La/Yb and Sm/Yb) for the Pohnpei island and the Louisville seamount lavas in every 5 km of lithosphere thickness. The basalts with MgO content of > 6 wt% for Louisville seamounts and > 9 wt% for Pohnpei Island are considered. In reviewed manuscript, (La/Yb)N and (Sm/Yb)N show statistically positive correlation with lithosphere thickness(Fig. 8 in manuscript). This supports a first-order control of lithosphere thickness on the chemical composition of ocean island lavas.
- The manuscript relies heavily on major element modeling undertaken with Petrolog, however there is little attempt to use the trace element and in situ data (the latter is not considered at all in the Discussion) to add a check on the modeling. For example, the authors could use REE compositions to calculate the extent of melting in a garnet-bearing source and compare this to the results shown in Figure 7. Or, the authors could compare their in situ data to the mineral composition outputs of Petrolog to verify that their selected initial conditions are producing mineral compositions similar to what they actually measure (c.f., Herzberg et al., 2014 EPSL, although in this paper Petrolog is not used in favor of the authors’ own calculations). Calculated LLD could also consider the effects of crustal and lithospheric assimilation on magma compositions; such assimilation has been shown to exert strong controls on trace element compositions in other hotspots (c.f., Class & Goldstein, 1997 EPSL). If the authors want to further try to connect their data to lithospheric thickness, they could test whether there are differences in “canonical” trace element ratios between areas with different lithospheric thicknesses to see if different depths of melting induce differences in the relative abundance of enriched/fusible mantle lithologies.
Response: Thanks for the hepful suggestions. We have tried to calculate the extent of mantle melting in a garnet-bearing source using REE compositions such as La/Yb, La/Sm and Sm/Yb ratios, the distribution coefficients of REE significantly affect the final calculation results. The results calculated using REE is higher than those by major element (Fig. 2, See attached pdf file).
We compared in-situ data of olivine and clinopyroxene to the mineral composition outputs of Petrolog3. Only SiO2, FeO and MgO content are given for olivine and SiO2, Al2O3, FeO, MgO and CaO content are given for clinopyroxene. The SiO2 and FeO content of olivine are well match with the olivine composition outputs of Petrolog3. The Al2O3, FeO content of clinopyroxene are slightly higher than clinopyroxene composition outputs of Petrolog3, but the CaO content much lower than clinopyroxene composition outputs of Petrolog3 (Fig. 3, See attached pdf file).
In addition, lithospheric mantle can affect magma evolution from source to eruption, involving melt-lithosphere interaction and fractional crystallization in the crust, We therefore speculate that the composition of Pohnpei Island lavas, to some extent, was modified when penetrating through the thick lithosphere, however, the effects of crustal and lithospheric assimilation on magma compositions was not evaluated here.
In summary, I’m left with the feeling that that paper underutilized the reported data, and instead tried to focus on correlations between geochemical and geophysical data that are generally difficult to definitively establish. I don’t think the authors’ ideas are bad ones, just not well supported by the available data. If the authors want to keep the direction of the paper, they should be clearer about the limitations of their data and how they support their hypotheses.
I’ve also left a number of detailed comments in the manuscript PDF for the authors to review.
Response: Thanks. This has been done and marked in reviewed manuscript. Some specific comments are answered in the attached pdf file of the first manuscript.

Round 2
Reviewer 1 Report
The manuscript has been revised according to the suggestion, hence, I recommend it for publication in Minerals.
Only one suggestion. Caption of Fig. 1. is incomplete. Please add a sentence about the inset in that modified figure and explain the abbreviations shown there.
Reviewer 1
Author Response
The manuscript has been revised according to the suggestion, hence, I recommend it for publication in Minerals.
Only one suggestion. Caption of Fig. 1. is incomplete. Please add a sentence about the inset in that modified figure and explain the abbreviations shown there.
Response: Thanks. This has been done.